# Health Patterns among Migrant and Non-Migrant Middle- and Older-Aged Individuals in Europe—Analyses Based on Share 2004–2017

**DOI:** 10.3390/ijerph182212047

**Published:** 2021-11-16

**Authors:** Nico Vonneilich, Daniel Bremer, Olaf von dem Knesebeck, Daniel Lüdecke

**Affiliations:** 1Institute of Medical Sociology, University Medical Center Hamburg-Eppendorf, 20246 Hamburg, Germany; o.knesebeck@uke.de (O.v.d.K.); d.luedecke@uke.de (D.L.); 2Department of Medical Psychology & Center for Health Care Research, University Medical Center Hamburg-Eppendorf, 20246 Hamburg, Germany; da.bremer@uke.de

**Keywords:** depression, self-rated health, functional limitations, older age, migrant status, health inequalities, trend analysis, Europe

## Abstract

Introduction: European populations are becoming older and more diverse. Little is known about the health differences between the migrant and non-migrant elderly in Europe. The aim of this paper was to analyse changes in the health patterns of middle- and older-aged migrant and non-migrant populations in Europe from 2004 to 2017, with a specific focus on differences in age and gender. We analysed changes in the health patterns of older migrants and non-migrants in European countries from 2004 to 2017. Method: Based on data from the Survey of Health, Ageing and Retirement in Europe (6 waves; 2004–2017; *n* = 233,117) we analysed three health indicators (physical functioning, depressive symptoms, and self-rated health). Logistic regression models for complex samples were calculated. Interaction terms (wave * migrant * gender * age) were used to analyse gender and age differences and the change over time. Results: Middle- and older-aged migrants in Europe showed significantly higher rates of depressive symptoms, lower self-rated health, and a higher proportion of limitations on general activities compared to non-migrants. However, different time trends were observed. An increasing health gap was identified in the physical functioning of older males. Narrowing health gaps over time were observed in women. Discussion: An increasing health gap in physical functioning in men is evidence of cumulative disadvantage. In women, evidence points towards the hypothesis of aging-as-leveler. These different results highlight the need for specific interventions focused on healthy ageing in elderly migrant men.

## 1. Introduction

In the last decades, European countries have witnessed immigration flows, leading to more diverse populations in Europe [1]. The share of migrant populations in European countries is rising; over the last thirty years it rose from around 7% to about 12% in 2020, with considerable variation between countries [2].

Early research detected the health advantages of first-generation immigrants [3], who initially showed a better health status than non-migrants in host countries [4,5]. Health selection in the countries of origin, e.g., for labour immigration, and the process of migration itself, which often requires good health, played an important role in this. But the health of migrants deteriorates as they settle in host-countries [6], due to socioeconomic disadvantages, increased risks in the workplace, or even experiences of discrimination, among other factors. Migrant and non-migrant populations face different health risks in European countries over the course of their lives [7,8,9].

Given the ongoing population ageing and increasing longevity in many European countries [1,10], it is necessary to focus on health differences between middle- and older-aged migrant and non-migrant populations, aged 50 years and older. Studies indicate poorer self-rated health, higher rates of depression, worse physical functioning, and lower life expectancy in older migrant populations [11,12,13,14]. However, most of these studies do not observe trends over time, nor do they consider variations regarding gender or age. When looking at health patterns in migrant and non-migrant elderly, a steeper rate of health decline was observed in older migrants [13].

When considering the health disparities between older migrant and non-migrant populations and their development over time, different hypotheses have been discussed: aging-as-leveler, persistent inequality, and cumulative disadvantage. The aging-as-leveler hypothesis states that health differences between population groups decrease in late life, when physical and mental limitations increase in all groups. The persistent inequality hypothesis states that, regardless of age, inequalities remain stable. And the hypothesis of cumulative disadvantage states that across the lifespan, socioeconomic disadvantages cumulate and lead to an increase in health risks and health inequalities between migrant and non-migrant elderly populations [15]. Evidence exists for the accumulation of disadvantage [13,16,17], for aging-as-leveler [18], and for persistent inequality [19], with most of the research focusing on black, Hispanic and non-Hispanic white populations in the US.

Earlier research analysing trends over time in the health of migrant and non-migrant elderly has often focused on the health of migrant and non-migrant populations in general, without specifying age or gender [13,14]. As migration background is one among other relevant factors associated with health, such as gender, age, or socioeconomic status [5,20], these factors are taken into account in the present study. Earlier studies indicated gender differences in health disparities according to migration [20,21,22].

Based on SHARE data, the health patterns of migrant and non-migrant middle- and older-aged individuals in different European countries were investigated. Moreover, time trends over 14 years were examined. Three health outcomes were assessed, as differences between groups might vary depending on the indicator used and in order to provide a more holistic and complex picture of health [15]: physical functioning, depression, and self-rated health (SRH). Therefore, the aim of the present trend analysis was to analyse health patterns in migrant and non-migrant middle- and older-aged individuals in Europe, focusing on individuals aged 50 years and older over a time period of 14 years, with a specific focus on gender and age differences.

## 2. Methods

### 2.1. Data

The analyses were based on data from SHARE, the Survey of Health, Ageing, and Retirement in Europe [23,24,25,26,27,28]. The sample comprised 233,117 observations from 28 European countries and Israel across 6 out of 7 waves (2004, 2006, 2010, 2013, 2015, and 2017), including Israel (Austria, Belgium, Bulgaria, Croatia, Cyprus, Czech Republic, Denmark, Estonia, Finland, France, Germany, Greece, Hungary, Ireland, Israel, Italy, Latvia, Lithuania, Luxembourg, Malta, Netherlands, Poland, Portugal, Romania, Slovakia, Slovenia, Spain, Sweden, and Switzerland). Data from the third wave were excluded, since this was a retrospective SHARELIFE survey. Based on the population registers, SHARE used probability samples within the countries and included non-institutionalized adults aged 50 years or older and, if available, their partners. Further exclusion criteria were being incarcerated, moved abroad, unable to speak the language of questionnaire, deceased, hospitalized, moved to an unknown address, or not residing at the sampled address [29,30].

### 2.2. Measures

The health patterns were assessed using three indicators that were dichotomised for analytical reasons. Physical functioning was measured using the global activity limitation index (GALI) [31]. The goal of this index is to assess perceptions in both general and more specific populations regarding long-standing, health-related limitations in common daily activities [31]. The GALI was measured by asking: “For the past 6 months at least, to what extent have you been limited because of a health problem in activities people usually do?”. Possible answers were “severely limited”, “limited, but not severely” and “not limited”. The first two limitations were combined for the binary variable with the categories 0 (“not limited”) and 1 (“limited”).

As a measure of mental health, the EURO-D was used to assess depressive symptoms [32]. The EURO-D was developed to allow the cross-country measurement and comparison of depressive symptoms. It includes 12 relevant symptoms of depression. The score ranged from 0 (“not depressed”) to 12 (“very depressed”). A binary variable for “depression” was categorised as 0 (“not depressed”) if the EURO-D score was lower than 4, otherwise it was categorised as 1 (“case of depression”), as proposed by the authors of the EURO-D scale [32].

The SRH was measured by asking, “Would you say your health is…”, offering five options as potential answers (“very good”, “good”, “fair”, “bad” and “very bad”). The variable was dichotomised, with 0 indicating “good or very good health”, and 1 “less than good health”.

Migrant background, age, and gender were used as the main predictors. The respondents’ age and gender were recorded at each wave. The respondents were considered as migrants if they were not born in the country of data collection. Wave is a continuous variable, ranging from 1 to 7, reflecting the respective year of data collection, where “1” relates to the year 2004, while “7” stands for 2014.

The household size and educational level of the respondents were used as further covariates. Educational levels were based on the International Standard Classification of Education [33] and recoded into three levels: “low (lower/upper secondary)”, “mid (post-secondary)” and “high (tertiary)”.

### 2.3. Analyses

For each of the three dichotomised health indicators, a logistic regression model for complex samples was calculated, using quasi-binomial links to properly account for survey-weighting, disproportional sampling, and selective mortality. The country variable was used to define the strata in the survey design; hence, the regression models accounted for the fact that the respondents were clustered within different countries. Robust Horvitz–Thompson standard errors were reported [34]. Models without interaction terms were calculated first, to investigate the overall associations between the dependent variables and the predictors of interest, migration, gender and age. The interactions between wave, migration, gender, and age were analysed to explore changes over time and to stratify the results by the aforementioned variables (migration, gender, and age). Age was standardized (z-score). Predictors that vary over time, particularly age, usually exert an effect within subjects (e.g., the individual outcome level changes when a person gets older) and between subjects (e.g., outcome levels differ between younger and older people). This results in biased regression coefficients, because both the within- and between-effect are captured in *one* coefficient. This is called “heterogeneity bias” [35]. To avoid this bias, age was separated into its within- and between-component. Since differences between age groups were the main focus of interest, the between-effect of age was used in the interactions. Educational status, household size, and the within-effect of age were included as further covariates. The associations between the interaction of the focal predictors and the three indicator variables are presented as predicted probabilities. The trends for each combination of the focal predictors’ categories (migrant yes/no, female/male gender, and middle/older-aged persons) are shown, resulting in figures with four panels per health indicator. Since age was a continuous variable, the values at one standard deviation below and above the average age in the sample were chosen as representative values through which to distinguish between younger and older respondents, following the suggestions from Aiken and West [36]. Thus, the figures represent model predictions as if an individual was at the age of 60 or 80, respectively. To answer the question of whether the change over time for both migrants and non-migrants and whether the comparison between these trends was statistically significant, contrast analyses and pairwise comparisons for the predicted probabilities were conducted to test for the differences between the stratified groups. The related *p*-values are shown in the figures. All the analyses were performed using the R statistical package (R Core Team, Vienna, Austria) [37], including the packages “survey” [38], “ggeffects” [39] and “emmeans” [40].

## 3. Results

Our analyses were based on data from 233,117 cases (55% female, 45% male, see Table 1). Overall, 10.4% were classified as migrants (10,923 men, 13,351 women). In the sample, 39.3% of the respondents were 70 years or older. The mean age of the non-migrants was 67.0 years and in migrants it was 66.6 years. The following significant differences between migrant and non-migrant populations were observed: the non-migrant populations were older (higher share of people aged 70 years and older) and had a higher share of lower education (45% upper secondary level or lower). Migrants across all waves under study reported a significantly higher share of GALI and a significantly higher share of poor SRH.

### 3.1. Associations between Health Indicators, Age, Gender and Migrant Status

In model 1 (Table 2), the regression models showed a 24% increased risk of reporting limited GALI for migrant populations, after controlling for wave, age (between and within subjects), gender, educational status, and household size. The age between and gender were not significantly associated with limited GALI. Similar results were found for depression (model 2). The migrant populations demonstrated a 19% increased risk for depressive symptoms compared to non-migrants. Model 3 demonstrates that migrants had a 28% higher chance of reporting poor SRH.

### 3.2. Subgroup Analyses and Time Trends

To analyse the interactions between the three focal variables (migrant background, age, and gender) with regard to three health indicators (limited GALI, depression, and poorer SRH), three logistic regressions were conducted, including several interaction terms (see Appendix A).

#### 3.2.1. Global Activity Limitation Index

Figure 1 presents the predicted probabilities of respondents who were limited in daily activities for migrant background, age, and gender over time in four line graphs (Figure 1a–d). The left-handed graphs show middle-aged males (a) and older-aged males (b). The right-handed graphs show middle-aged females (c) and older-aged females (d). When looking at the predicted probability models for GALI across the six waves, it can be observed that significantly more middle-aged migrant men reported limitations in global activities than non-migrant men, but that these differences decreased over time (Figure 1a). In terms of older males (on average 80 years of age), the trends in GALI over time contrasted significantly, as a higher proportion of migrant men reporting limitations in global activities over time was observed (Figure 1b). A significant increase in the predicted probability of GALI over time was found in the non-migrant men. In females, different trends regarding GALI over time were found. The gap between migrant and non-migrant elderly women narrowed both in the middle- and in the older-age groups (Figure 1c,d). In both age groups, GALI in non-migrant women significantly increased over time, leading to a reduction in the health gap in GALI observed in the early waves of SHARE. Generally, no significant trends were found for pairwise comparisons of trends in patterns of GALI over time between the migrant and non-migrant populations.

#### 3.2.2. Depression

Figure 2 presents the predicted probabilities of the respondents who suffered from depression on migrant background, age, and gender (Figure 2a–d).

In depression, a significant increase over time was found in non-migrant middle-aged men (Figure 2a). For migrant men, a stagnation over time among the middle-aged and a decrease among the older-aged was identified, although the latter did not show a significant level (Figure 2b). A significant increase in depression over time was also found in middle-aged non-migrant women, but not for migrant women in the same age group (Figure 2c). For middle-aged men and women alike, the gap between migrant and non-migrant populations in depression found in the early waves disappeared over time. In older-aged individuals, the likelihood of depressive symptoms increased over time, particularly in non-migrant men (Figure 2b,d). No significant trends for pairwise comparisons of patterns in depression over time between migrant and non-migrant elderly were identified.

#### 3.2.3. Self-Rated Health

Figure 3 presents the predicted probabilities of respondents who reported poorer SRH for the migrant and non-migrant elderly, distinguished by age and gender (Figure 3a–d).

In men, the gap in SRH between migrant and non-migrant populations increased over time (Figure 3a,b). This trend was particularly pronounced among middle-aged males, reaching significant differences in wave 4 and 5 (2010 and 2013, Figure 3a). A slightly significant increase in probability of reporting poor SRH over time was found among older-aged non-migrant men only. In women, a contrary time trend was observed (Figure 3c,d). In middle-aged females, a significant gap in wave 1 (2004), with higher risks of reporting bad SRH in migrant populations, diminished over time (Figure 3c). No significant differences and very similar amounts of poorer SRH were found in wave 7 (2017). An increase in time of poorer SRH in non-migrant middle-aged women and a decrease in migrant middle-aged women was found (Figure 3c). For older-aged women, the gap was narrowing as time trends of poor SRH were increasing in non-migrant populations (Figure 3d). This increase was more pronounced than in migrant older-aged women, leading to a decrease of the initial gap in SRH. No significant trends for pairwise comparisons in patterns of SRH over time between migrant and non-migrant elderly were identified.

## 4. Discussion

### 4.1. Summary

Based on the trend analysis of the data from SHARE for 2004 to 2017, this study shows different health patterns between middle and older-aged migrant and non-migrant-populations. A significant health gap for migrant populations exists, with higher risks for reporting worse health outcomes in all three indicators (physical functioning, depressive symptoms, and SRH) for migrant populations. Over time, these gaps develop differently, especially when age and gender are taken into account. In women, the health gap between migrant and non-migrant elderly narrowed significantly across all three health indicators. Moreover, interestingly, there are indications for a reverse health gap, with higher risks of reporting depression in non-migrant elderly women. In men, there is evidence for widening health gaps, such as the SRH in middle-aged men and physical limitations in older-aged men. In depression, differences between migrant and non-migrant men in the all ages under study diminished over the years. The results presented need to be interpreted with caution, as the differences in time trends between migrant and non-migrant populations did not reach significant levels.

While previous studies found some evidence for deteriorating health trends over time, especially in migrant populations [13,41], the analysis revealed a more complex picture of health patterns and trends over time in the health of migrant and non-migrant elderly in Europe by considering age and gender differences (see also [5]). The results show that health differences between the migrant and non-migrant elderly change over time. There is no clear trend of deteriorating health among migrants in Europe [13]. Evidence indicates a widening of health disparities among men, while among women there is evidence indicating a reduction in these disparities.

With regard to theoretical considerations of the development of health disparities over time [15], some results point towards the hypothesis of accumulation. This is especially true for men with regard to physical limitations in older age groups and for SRH. When looking at the health trends in women, there is evidence for the hypothesis of aging-as-leveler, since the health gap between migrant and non-migrant elderly women decreased over time, especially in GALI and SRH. As other scholars have pointed out, downplaying differences among elderly migrant populations such as age or gender will likely lead to false assumptions regarding the development of health patterns [5,41]. The findings presented here underline these conclusions.

How can these age and gender differences in health patterns be explained? In terms of increasing health gaps in GALI in elderly men, physically demanding jobs and higher risks in the workplace experienced during the lifespan might play an important role [4,42,43], especially for the older cohort of elderly migrant men. Elderly migrant women might be less likely to suffer from physical risks in the workplace. Health gaps in SRH and depression suggest psychosocial explanations, such as social isolation and the experience of discrimination [8,43], and some research suggests that psychological distress in migrant populations might increase with age [44]. In order to explain differences in health patterns between men and women, other scholars have pointed towards the role of welfare states and other contextual factors [45,46], such as health systems, health coverage or health equity concerns in national strategies, which are likely to affect health services and access to health care for certain (marginalised) groups (e.g., due to language barriers) as well as living and working conditions of women with corresponding consequences for health in later life stages.

We find evidence for higher health risks among migrant elderly despite their higher educational status. How can this be explained? Educational status might not reflect the true socioeconomic status of migrant elderly, as the process of migration often occurs after the end of educational training. In the process of migration, recognition of educational attainment is difficult, resulting in a greater likelihood of lower-qualified occupations and, consequently, lower income [47]. It has also been shown that within Europe, tertiary level graduates are more likely to migrate, which is in line with the finding of higher educational attainment within the migrant populations studied here [48].

### 4.2. Limitations

Some methodological limitations need to be taken into account when interpreting the results. Firstly, SHARE excludes institutionalized women and men [49]. The survey is representative for community-dwelling people 50 years and older with regard to the participating European countries [49]. Since institutionalized individuals show worse health patterns, this could have led to more positive health trends in the analyses. Furthermore, panel attrition of potentially older and more handicapped people can contribute to sample selection, limiting the representativeness of the data and the generalisability of results [29].

Moreover, SHARE includes only those participants that were able to speak the official language of the respective country [49]. Consequently, migrants with weaker language skills are underrepresented in the sample [13]. This effect might partially explain the higher educational status within the migrant populations under study. This could also be also true for specific migrant populations, e.g., undocumented migrants in precarious living conditions and asylum seekers, among others. Conceivably, this limitation would make an underestimation of health differences more likely. Additionally, the SHARE data did not provide detailed information on the migrant background, such as length of stay, migration history, or the nationality of the migrants’ parents. When looking at migrant subpopulations, the design of the data is improvable [13]. Consequently, the variable of migrant background represents no more than a proxy for the diversity of migrant experiences [4]. In particular, since the proportion of migrants is comparably low in the overall sample, a further differentiation of migrant populations (such as country of origin, length of stay, or type of immigration) would lead to small subsamples and would not be feasible for the present analysis.

The analyses are to some extent crude as a limited set of control variables, namely age, gender, household size, and education were included, in order to reduce complexity and enhance the interpretation of the results. Other potential explanatory variables were not included as the focus of the present analysis is to analyse changes over time rather than to explain the complex associations between the variables.

Our three health indicators are characterized by self-reporting. GALI and EURO-D are validated and widely used instruments [50]. On the one hand, SRH is based on a general single item and caution is recommended when drawing conclusions from studies using SRH as health indicator [51]. On the other hand, SRH represents a proper summary of health status [52] and it has been shown to be a valid measure of health status, especially among older adults, regardless of varying cultures and social conditions [53,54,55]. The three health indicators were dichotomised for reasons of comparability and clearness. As a consequence, the analyses are crude to some extent.

Lastly, all the analyses are based on the pooled data of 28 European countries and Israel. This procedure involves the risk of missing country-specific trends caused by different health systems [56], migration patterns [57], and welfare regimes [58]. In this regard, country-specific health patterns in migrant and non-migrant populations were analyzed but did not reveal any clear country-specific patterns (details not shown here).

## 5. Conclusions

The results of the present analyses highlight the need for interventions focused on healthy ageing for the migrant elderly, with a specific focus on middle- and older-aged migrant men. Especially in SRH and GALI, widening health gaps between non-migrant and migrant elderly men were observed. Migration should be included in healthy ageing policies in Europe [20,58]. Policies for migrant integration can help to reduce health disparities [59]; explicit migrant health policies are needed in all European countries in order to adapt health systems to the specific risks and needs of migrant populations [4]. These policies and interventions should be targeted regarding age and gender of migrants. As the results show, treating migrants as a homogenous group underestimates the differences in health and health patterns over time. Further migrant and health specific panel data are needed. Further differentiation of migrant populations in future studies is needed, for example by oversampling migrant populations, as they could take countries or regions of origin and length of stay in host countries as further potential explanations of health differences into account [4,41]. This also implies greater efforts to include migrant populations, especially those who are potentially harder to reach, in future research and to reduce existing barriers, such as language restrictions.

## Figures and Tables

**Figure 1 ijerph-18-12047-f001:**
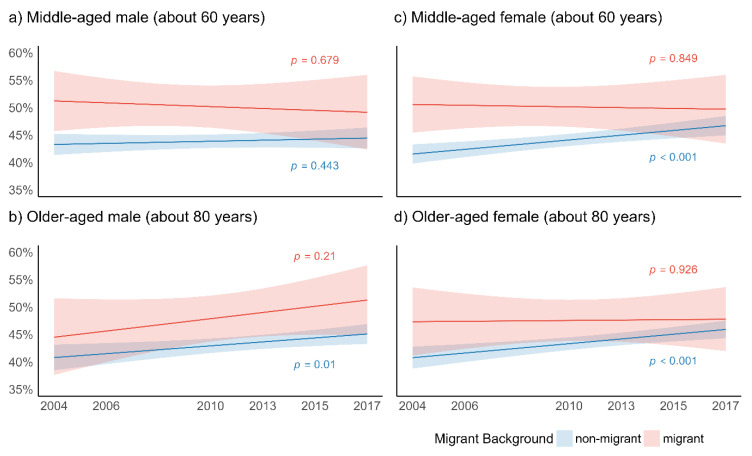
Predicted probabilities of Global Activity Limitation Index (% limited in daily activities) for migrant background, age, and gender (across six waves of SHARE); *p*-values relate to change over time for migrant and non-migrant elderly.

**Figure 2 ijerph-18-12047-f002:**
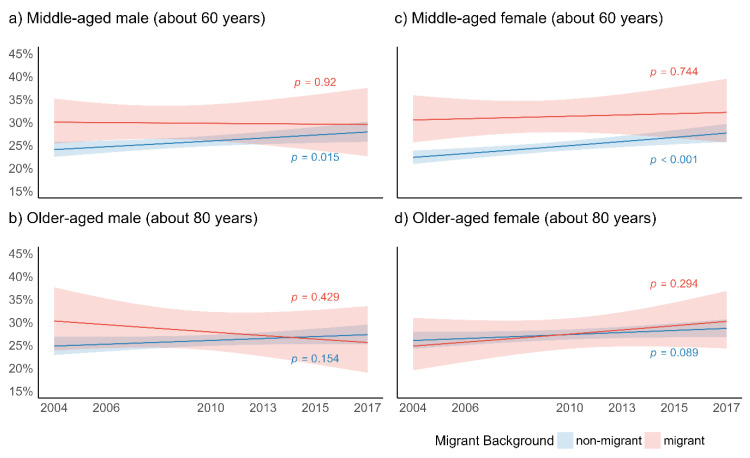
Predicted probabilities of EURO-D (% depressed = four or more symptoms of depression) for migrant background, age and gender (across six waves of SHARE); *p*-values relate to change over time for migrant and non-migrant elderly.

**Figure 3 ijerph-18-12047-f003:**
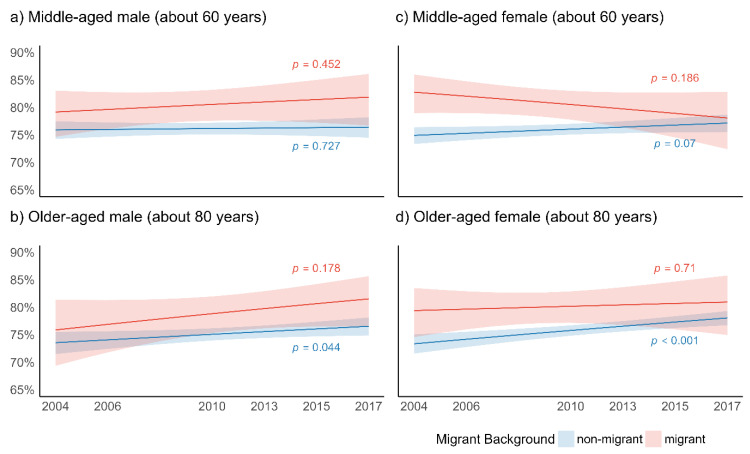
Predicted probabilities of self-rated health (% less than good health) for migrant background, age, and gender (across 6 waves of SHARE; model 3.2); *p*-values relate to change over time for migrant and non-migrant elderly.

**Table 1 ijerph-18-12047-t001:** Descriptive statistics of SHARE data (*n* = 233,117; across all countries and waves).

	Wave 1 (2004)		Wave 2 (2006)		Wave 4 (2010)		Wave 5 (2013)		Wave 6 (2015)		Wave 7 (2017)		Total		*p*-Value for Difference
Variable	Non-Migrants (*n* = 19,585)	Migrants (*n* = 2554)	Non-Migrants (*n* = 21,241)	Migrants (*n* = 2334)	Non-Migrants (*n* = 37,043)	Migrants (*n* = 4136)	Non-Migrants (*n* = 40,535)	Migrants (*n* = 5452)	Non-Migrants (*n* = 40,975)	Migrants (*n* = 4955)	Non-Migrants (*n* = 49,464)	Migrants (*n* = 4843)	Non-Migrants (all waves, *n* = 208,843)	Migrants (all waves, *n* = 24,274)	
Mean Age (SD)	64.8 (10.4)	64.9 (10.2)	64.8 (10.4)	64.3 (10.4)	65.7 (10.5)	65.4 (9.7)	67.8 (10.5)	67.6 (10.3)	68.4 (10.8)	68.0 (10.2)	69.5 (10.4)	69.7 (10.0)	67.0 (10.7)	66.6 (10.3)	0.007 ^a^
Age (70 and older), %	31.9	32.0	31.4	27.7	34.9	32.2	40.3	40.0	42.8	40.1	46.1	48.5	38.5	36.9	0.013 ^b^
Female Gender, %	53.9	54.0	53.5	56.5	55.7	52.5	55.6	53.7	55.9	58.2	57.3	57.5	55.5	55.4	0.941 ^b^
Mean Household Size (SD)	2.2 (1.0)	2.2 (1.0)	2.1 (1.0)	2.1 (0.9)	2.2 (1.0)	2.1 (0.9)	2.1 (1.0)	2.1 (0.9)	2.1 (1.0)	2.0 (1.0)	2.0 (1.0)	1.90 (1.0)	2.1 (1.0)	2.1 (1.0)	0.279 ^c^
Education															<0.001 ^b^
lower/upper secondary, %	53.3	38.5	48.9	37.6	45.8	34.5	43.2	33.8	42.2	32.9	39.7	30.2	45.0	34.6	
post-secondary, %	30.7	37.6	34.7	39.1	37.0	40.3	36.9	36.9	38.8	38.6	41.1	39.9	36.9	38.7	
tertiary, %	16.0	24.0	16.4	23.3	17.2	25.2	19.9	29.3	19.0	28.5	19.2	29.8	18.1	26.7	
Health Indicators															
Global Activity Limitation Index, % limited	43.7	51.6	44.9	48.7	49.0	51.3	45.2	47.4	45.9	49.2	47.5	51.0	46.2	49.9	<0.001 ^b^
Depression, % depressed	27.6	32.1	27.9	28.4	30.5	30.6	28.4	30.0	29.6	31.6	31.4	29.8	29.0	30.5	0.074 ^b^
Self-Rated Health, % less than good	76.3	79.4	78.6	81.7	79.5	80.8	76.3	79.8	77.3	78.1	79.0	80.7	77.9	80.1	0.004 ^b^

^a^ Design-based *t*-test; ^b^ Pearson’s Chi^2^ (Rao & Scott adjustment); ^c^ Design-based Kruskal–Wallis test.

**Table 2 ijerph-18-12047-t002:** Logistic regression models with survey design for GALI (model 1), depression (model 2) and self-rated health (model 3) (SHARE, waves 1, 2, 4–7, *n*= 233,117); odds ratios, 95% confidence intervals (CI) and significances (*p*).

Parameter	Limited Global Activity Limitation Index (GALI) (Model 1)			Depression (Model 2)			Poorer Self-Rated Health (Model 3)		
	Odds Ratio	95% CI	*p*	Odds Ratio	95% CI	*p*	Odds Ratio	95% CI	*p*
(Intercept)	0.95	(0.89, 1.01)	0.088	0.53	(0.50, 0.57)	<0.001	4.85	(4.51, 5.21)	<0.001
Wave	1.03	(1.02, 1.03)	<0.001	1.03	(1.02, 1.04)	<0.001	1.02	(1.01, 1.03)	<0.001
Migrant (ref. non-migrant)	1.24	(1.15, 1.34)	<0.001	1.19	(1.09, 1.29)	<0.001	1.28	(1.17, 1.40)	<0.001
Age (between subjects)	0.98	(0.95, 1.01)	0.184	1.03	(1.00, 1.07)	0.086	0.98	(0.95, 1.02)	0.359
Female Gender (ref. male)	1.01	(0.98, 1.05)	0.367	1.00	(0.97, 1.04)	0.863	1.02	(0.98, 1.06)	0.256
Education (ref. lower/upper secondary)									
post-secondary	0.73	(0.70, 0.77)	<0.001	0.57	(0.54, 0.60)	<0.001	0.60	(0.57, 0.64)	<0.001
tertiary	0.52	(0.49, 0.55)	<0.001	0.40	(0.38, 0.43)	<0.001	0.33	(0.31, 0.35)	<0.001
Household Size	1.00	(0.99, 1.02)	0.656	0.98	(0.96, 1.00)	0.055	1.00	(0.98, 1.02)	0.893
Age (within subject)	1.02	(1.00, 1.04)	0.020	0.99	(0.97, 1.02)	0.539	1.02	(1.00, 1.04)	0.080

## Data Availability

The SHARE data is available for research purpose after registering at the SHARE website (www.share-project.org, accessed on 14 November 2021).

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
