# Peer review of "Health Patterns among Migrant and Non-Migrant Middle- and Older-Aged Individuals in Europe—Analyses Based on Share 2004–2017"

_ijerph, 2021, doi:10.3390/ijerph182212047_

Round 1

Reviewer 1 Report

Many thanks for the opportunity to review your article.

The paper is well written, well structured, and easy to read and follow.

The method used was appropriate for the study. I particularly like how the researchers discussed the biases and what they’ve done to avoid or limit the biases as well as testing other indicators to identify health patterns by country specific.

Discussion section clearly summarised and discussed the findings including the gap and limitations, citing previous research in the same field to prove the outcomes.

Conclusion section directly highlight the key findings i.e., the need for interventions of healthy ageing in migrants’ elderly as well as acknowledging the limitations and the relevance of the study without repeating what was discussed in the discussion section.

Please consider the following suggestions:

  • Clearly state the aims of the paper (reframe the sentence between lines #11 & 12).
  • Avoid using the first-person pronoun i.e., “we” and ‘our” because this is a research paper. Review the whole article.
  • Define GALI and EURO-D not just naming the abbreviation. Research method and instruments are integral part of the research paper. Therefore, it is important to clearly describe or define the methods and measurements used.
  • Line #33 – name the migrant groups being referred here if available.
  • There’s a recent paper published that is relevant to your paper that you may wished to have a look.
    • Maskileyson, D., Seddig, D., & Davidov, E. (2021). The EURO-D Measure of Depressive Symptoms in the Aging Population: Comparability Across European Countries and Israel. Frontiers in Political Science, 3, 90.

Reviewer 2 Report

The paper is nicely written and relevant at the present time. The changes in time are the most interesting part. I have some remarks that needs to be taken into account when revising the paper.

Title

  • the definition of elderly meaning over 50 years gives a false idea to the readers. WHO seems to use 60 or 65 years. I suggest using over 50 years also in the title or middle- and older-aged as the authors use in Conclusions.
  • what does cumulative disadvantage mean here? Do they mean that several health variables are related to migration? In that case, it is better to be called cumulative health disadvantage because ‘disadvantage alone’ suggests a cumulation of socioeconomic disadvantages.

Abstract

  • what ”older” means? Mention the age of the cohorts and what means older.
  • the last sentence of interventions is better to be deleted because “interventions on healthy ageing” in general is self-evident and the results do not support that interventions on elderly men are effective anymore but should be at earlier ages. And interventions are not mentioned in conclusions, either.

Methods

  • The authors have used between-within component with age. But also, education varies in the population in the similar way when people are aging and when time passes. And I suppose similar changes concern migrant population, too. Should this be considered.
  • There were several countries in the sample which means that respondents cluster within countries. Why the authors did not use multilevel (hierarchical) models which seems to be best practices today?

Results

  • It is not possible to read Table 2 without going to the text. The legend must be revised so that it is readable without text.
  • In figures 1-2 the small figure descriptions, e.g., younger aged male (at 60 years) is difficult. I suggest using the real ages instead younger and older and exact ages. These are not 60 years only. Otherwise, the figures are nice and clear.

Discussion

  • 270-274. What are these welfare state and contextual factors that the authors refer to? There could be more explanations concerning the differences between migrants and non-migrants (r.263- ). What about access to health care and language barriers as well as cultural beliefs of health and origin of diseases connected to use of health care?

Reviewer 3 Report

Major comments

  1. I suggest you to run separate estimations for women and men, rather than mixing genders and using very complex interaction terms which are very hard to interpret. People are such different in the health context, that you should analyze them separately. It seems that you also have enough observations to run separate estimations by gender.
  2. It is not clear how the variable wave is used in the study. This will help the reader understand if its use to capture the trend is appropriate or not.  
  3. It is not clear whether you are using country fixed effects in your estimations. This is crucial when estimating pooled datasets because each country has a specific context that should be controlled for. For example it could control for the different health care systems of each country, or for different average health levels or even for the different types of immigration in terms of composition of nationalities countries have.
  4. They use a very limited set of controls. Immigrants and natives may differ in a wider set of characteristics that influence the health outcomes such as income for example. Other controls could be marital status, nr. children. Discuss this possibility also challenging it with other similar studies comparing the health outcomes of  immigrants and natives.
  5. Are you clustering the standard errors at the country level? You may know that observations are not independent within each country. If you claim they are independent, please state clearly why do you think it is so.
  6. In case you have information on the origin country of immigrants, you could even explore this type of heterogeneity. I expect that immigrants of the same age and gender coming from poorer/ less developed countries may show different health issues that also depend on (e.g.) the origin country health system. This could be a value added of this research.  

Minor comments

- In Introduction, line 29 rephrase “migration flows”à immigration flows/immigrants flow or migration inflows.

- In the results, instead of reporting the OR and the confidence intervals in parenthesis, you could just interpret the OR. For example, in Table 2 model 1, you could say that the probability to have limited physical functioning (a low GALI) is 24 % higher for immigrants compared to natives. The expression “showed increased risk of reporting” is not clear to me and may create some confusion for the reader. You could do the same with the other coefficients.

- The first paragraph of the results showing the data characteristics could fit better in the data section (2.1). Indeed it is not a result, but just the summary statistics of the data you are using in the analysis.

Round 2

Reviewer 3 Report

Thank you for addressing all the comments.

Good luck with your further research.